# The Prediction of Cu(II) Adsorption Capacity of Modified Pomelo Peels Using the PSO-ANN Model

**DOI:** 10.3390/molecules28196957

**Published:** 2023-10-06

**Authors:** Mengqing Jiao, Johan Jacquemin, Ruixue Zhang, Nan Zhao, Honglai Liu

**Affiliations:** 1Hebei Key Laboratory of Green Development of Rock and Mineral Materials, Hebei GEO University, Shijiazhuang 050031, China; jmq15931164981@163.com (M.J.); zhangruixue77@163.com (R.Z.); 2Materials Science and Nano-Engineering MSN Department, Mohammed VI Polytechnic University, Lot 660-Hay Moulay Rachid, Ben Guerir 43150, Morocco; johan.jacquemin@um6p.ma; 3School of Chemistry and Molecular Engineering, East China University of Science and Technology, Shanghai 200237, China; hlliu@ecust.edu.cn

**Keywords:** adsorption capacity, artificial neural network, particle swarm optimization, Pearson correlation, cosine amplitude method, Garson equation

## Abstract

It is very well known that traditional artificial neural networks (ANNs) are prone to falling into local extremes when optimizing model parameters. Herein, to enhance the prediction performance of Cu(II) adsorption capacity, a particle swarm optimized artificial neural network (PSO-ANN) model was developed. Prior to predicting the Cu(II) adsorption capacity of modified pomelo peels (MPP), experimental data collected by our research group were used to build a consistent database. Then, a PSO-ANN model was established to enhance the model performance by optimizing the ANN’s weights and biases. Finally, the performances of the developed ANN and PSO-ANN models were deeply evaluated. The results of this investigation revealed that the proposed hybrid method did increase both the generalization ability and the accuracy of the predicted data of the Cu(II) adsorption capacity of MPPs when compared to the conventional ANN model. This PSO-ANN model thus offers an alternative methodology for optimizing the adsorption capacity prediction of heavy metals using agricultural waste biosorbents.

## 1. Introduction

Copper is a toxic metal commonly found in wastewater. Excess copper uptake into the body can lead to hair loss, headache, gastrointestinal dysfunction, and central nervous system diseases [1]. Therefore, the removal of copper ions from industrial effluents is crucial before their discharge into aquatic environments. Agricultural wastes, used as biosorbents, possess several advantages because they are abundant, cheap, and renewable [2]. Pomelo peels, a kind of agricultural waste, present a quite high biosorption potential, as they are primarily composed of cellulose, hemicellulose, lignin, and pectin. These components contain carboxylic groups and phenolic acid moieties, which contribute to metal binding. At present, the use of pomelo peels as biosorbents to remove pollutants is widely discussed in the literature [3,4,5,6]. In addition, the Cu(II) adsorption of chemically modified pomelo peels (MPP) has also been reported by our group in a previous study [7].

Modeling is an indispensable stage for any process that needs to be scaled-up from Lab to Fab, but it could be also viewed as an important engineering approach for the optimization of laboratory pre-screening processes [8]. Nevertheless, biosorption is a complex process, owing to the numerous parameters and different biosorption mechanisms involved. Hence, it is difficult to simulate the results of a batch biosorption process using a single conventional mathematical model. Machine learning involves the use of artificial intelligence to learn from given data and simulate complex systems. Some of the most common ML algorithms are the support vector machine (SVM) model, the random forest (RF) model, fuzzy logic (FL), and the artificial neural network (ANN) model [9]. Among them, the ANN model provides a superior nonlinear ability to learn from a variety of complicated systems. Several studies on the adsorption capacity of heavy metal ions, such as Pb(II), Ni(II), Co(II), Cr(VI), Cu(II), and Zn(II), based on the ANN model have been reported [7,10,11,12,13,14,15,16].

There have been few studies on the optimization of ANNs’ weights and biases using intelligent algorithms. Conventional ANN models are prone to falling into local extremes when solving complex nonlinear problems, and the adsorption capacity of a heavy metal is affected by numerous nonlinear factors. Therefore, the introduction of intelligent algorithms into ANN models can improve their predictive ability. The PSO algorithm is widely utilized to optimize model performance, owing to its higher convergence rate compared to other evolutionary algorithms [17]. The prediction is not too far off from the global optimum, even if the global optimum cannot be truly found. However, hybrid methods have not been utilized in the prediction of the Cu(II) adsorption capacity of MPP, to date.

Herein, a particle swarm optimized artificial neural network (PSO-ANN) model was developed to predict the Cu(II) adsorption capacity of MPP. Because PSO finds the optimal initial weights and biases, PSO-ANN can achieve a higher accuracy than the conventional ANN model. Therefore, a database was initially built, and a sensitivity analysis of the input variables was performed. Then, the ANN and PSO-ANN models were established by using this database. During this stage, the collected datasets were fed into each model for training and testing. Finally, the performances of both models were evaluated, compared, and analyzed statistically to highlight further recommendations.

## 2. Results and Discussion

The objective of this study was to predict the Cu(II) adsorption capacity of MPP by using ANN and PSO-ANN models. The following subsections provide descriptions of the constructed models and their prediction outcomes.

### 2.1. Results Predicted from the ANN Model

The number of hidden layer neurons was determined by using a trial-and-error method as equal to 9. As a result, the configuration of the neural network was set as 4-9-1 (Figure 1). The model parameters of the ANN are listed in Table 1.

Table 2 displays the assessment results of the ANN model performance. The *R*^2^ values for the training and testing sets were close to 0.99977 and 0.99987, respectively, which were generally satisfactory. The RMSEs of the ANN were 0.0217 for the training set and 0.0159 for the testing set, while the MAE values were close to 0.0140 and 0.0125, respectively. In conclusion, the common ANN model has some generalization potential and can somewhat anticipate unknown data.

### 2.2. Results Predicted from the PSO-ANN Model

The prediction results of the standard ANN model depend significantly on its weights and biases. Thus, PSO was used to optimize the initial weights and biases of the ANN model training and preliminarily narrow the search space. This allowed the ANN model to reach the global minimum faster in the following iterative training instead of being prone to local minima. For the PSO-ANN model, the particle swarm size, number of iterations, learning factors, particle position constraint, particle velocity constraint, and inertia factor were set. It should be mentioned that the specification of the investigated problem had a significant impact on the particle swarm size [18]. A particle swarm size of 20 was found to be optimal after testing particle swarm sizes from 15 to 30 in intervals of 5. Table 3 displays the parameter settings of the PSO-ANN for this investigation.

The results of the PSO-ANN model performance were also assessed and then compared to those observed when using the ANN model, as reported in Table 2. In the case of the PSO-ANN, the *R*^2^ values for the training and testing sets were 0.99998 and 0.99999, respectively. The RMSEs were 0.0081 for the training set and 0.0077 for the testing set. The MAE for the training set was 0.0067, whereas it was 0.0066 for the testing set.

The *R*^2^ values of the training and testing sets for the PSO-ANN model were higher than those for the standard ANN model (Table 2). In comparison to the conventional ANN model, the RMSE and MAE values of the training and testing sets for the PSO-ANN were lower by varying degrees. Therefore, the PSO algorithm introduced into the ANN model increased the accuracy of the predicted values.

### 2.3. Discussion

The weight–bias matrices for the ANN and PSO-ANN are provided in Appendix B. In this study, a neural network optimized using the particle swarm algorithm was developed to predict the ability of MPP to adsorb Cu(II). The results for both models are listed in Table 2 in the preceding subsections and discussed further below.

A regression analysis of the ANN and PSO-ANN is provided in Figure 2. The majority of the data points are spread on both sides of the fitted lines, demonstrating the good ability of both models to fit the data (Figure 2). PSO-ANN achieved a higher *R*^2^, indicating its better fitting ability compared to the classical ANN approach. This is also reflected in other indicator values such as the RMSE, MSE, and MAE (Table 2). The RMSE values of the PSO-ANN (0.0081, 0.0077, and 0.0080) were significantly lower than those of the ANN (0.0217, 0.0159, and 0.0207) for the training, testing, and all data, respectively. In addition, the PSO-ANN had a lower MAE value for each phase. Therefore, the introduction of PSO into the ANN model improved its prediction accuracy, as the testing data were not included in the model training. As shown in Figure 3, the errors between the predicted and actual values based on the ANN model were larger than those based on the PSO-ANN. This indicates that the predicted values using the PSO-ANN were closer to the actual values than those using the ANN, leading to the same conclusion. Figure 4 displays the gradient change in the iterative training for the ANN and PSO-ANN. It can be seen from this figure that the gradients of the PSO-ANN and ANN both decrease sharply in the first 30 epochs. Then, these gradients decrease much slower prior to becoming stable after 30 epochs. As shown in Figure 4, the decline curve of the PSO-ANN is smoother and converges to a higher accuracy than that observed with the ANN model. In fact, some fluctuations on the gradient curve of the ANN model are observed during its training process. In addition, the training of the ANN stops at 349 epochs, even if there is still a certain distance from the ideal error value, indicating that there is a local optimal solution problem using this approach.

Based on the PSO-ANN weight matrix, the Garson equation [19] was then adopted to evaluate the relative importance of the input variables. Figure 5 depicts the relative importance of the input variables derived using the Garson equation. For this analysis, one can see that the initial Cu(II) concentration was the most influential input variable, followed by the temperature. Such a conclusion is consistent with the results obtained by the CAM, and further verifies the validity of the calculated connection weights of the developed PSO-ANN model. Furthermore, based on this analysis, the initial pH was found to have the lowest influence on the Cu(II) adsorption capacity, while, within the CAM, it was observed that the adsorption time had the smallest impact. However, the relative importance of the adsorption time (23.9%) was close to that of the initial pH (20.3%), which is consistent with the similar values of *S_i_* calculated by the CAM for the adsorption time (0.9439) and the initial pH (0.9626).

In brief, the introduction of PSO into the ANN was productive and successful. By virtue of its optimization capabilities, the particle swarm algorithm enhanced the ANN’s weights and biases, increasing the model accuracy and decreasing the prediction errors. Additionally, the adaptive treatment of the inertia factor efficiently improved the PSO algorithm to achieve a more comprehensive optimization search to converge more quickly to the global optimum. AI hybrid models can integrate and combine the advantages of both selected single models, thus leading to a better accuracy and predictive capability.

To further validate the reliability of the proposed model, comparisons with other hybrid methods based on neural networks were made. Table 4 displays the performances of other ANN-based hybrid methods, demonstrating the superiority of the PSO-ANN approach. Hence, the proposed hybrid PSO-ANN model was validated for predicting the Cu(II) adsorption capacity of MPP. However, it was undeniable that the PSO-ANN took a bit more time to reach the optimal prediction, when compared to the time requested to run the single ANN model. This main drawback could be also optimized by reducing the training time of the PSO-ANN model, which is one of our objectives for future work on designing proper applications based on the optimized prediction of heavy metal absorption using AI hybrid models. 

## 3. Materials and Methods

### 3.1. Database Description and Variable Analysis

A trustworthy database is necessary for successfully training machine learning models. Otherwise, the training results cannot reflect the actual situation, which ultimately leads to model training failure. The procedures were proposed herein to build a database compiling the Cu(II) adsorption capacities using MPP as a proof of concept.

A sufficient database not only consists of a large amount of data, but also requests a comprehensive range of input and output variables. In this study, experimental data were collected from our previously reported work [7] to create such a database. The Cu(II) adsorption capacity of MPP is influenced by numerous experimental factors, such as the temperature, initial pH, adsorption time, and initial Cu(II) concentration. Therefore, different experimental factors should be identified to increase the capability of the network learning when collecting the adsorbed data. In total, a database composed of 100 experimental datasets was established in this study. These data were split into training (80) and testing (20) datasets. The database consisted of four input variables (i.e., temperature, initial pH, initial Cu(II) concentration, and adsorption time), while the output variable was the Cu(II) adsorption capacity of the MPP. A statistical analysis of the input and output variables of the database is provided in Table 5.

Notably, the interdependency between the input variables hinders the model’s performance due to multicollinearity [25,26]. Hence, the Pearson correlation coefficient (*r*) was used to perform a redundancy analysis for the input variables and is expressed as follows [27]:(1)$$r=\frac{\sum_{k=1}^{n}(x_{ak}-\bar{x_{a}})(x_{bk}-\bar{x_{b}})}{\sqrt{\sum_{k=1}^{n}{\left(x_{ak}-\bar{x_{a}}\right)}^{2}}\sqrt{\sum_{k=1}^{n}{\left(x_{bk}-\bar{x_{b}}\right)}^{2}}}$$
where *n* indicates the number of data; *x_a_* and *x_b_* are the input variables; and $\bar{x_{a}}$ and $\bar{x_{b}}$ are the mean values of the variables. Thus, *r* is between −1 and 1; the larger its absolute value, the stronger the linear correlation between the input variables. If 0.9 ≤ $\left|r\right|$ ≤ 1, the information redundancy between the two variables is high; only one variable is retained, and the other is deleted.

Figure 6 displays the heat map of the Pearson correlation coefficients for the input variables. The absolute values of *r* between the input variables were all less than 0.2, reflecting the impact of diminishing multiple collinearities.

### 3.2. Sensitivity Factor Analysis of Input Variables

A sensitivity factor analysis is an important method used to calculate the effect of the input variables on the output variable. Herein, the cosine amplitude method (CAM) was employed [28]. The sensitivity of each input variable can be expressed by:(2)$$S_{i}=\frac{\sum_{k=1}^{n}x_{ik}y_{k}}{\sqrt{\sum_{k=1}^{n}x_{ik}^{2}\sum_{k=1}^{n}y_{k}^{2}}}$$
where *x_i_* is the input variable, y is the output variable, and *S_i_* is the degree of influence of each input variable on the output variable. The values of *S_i_* between the input variables and the Cu(II) adsorption capacity are shown in Figure 7. The most significant influence on the Cu(II) adsorption capacity was the initial Cu(II) concentration, followed by the temperature, while the adsorption time had the lowest impact.

### 3.3. Methods

ANN models have been utilized widely because of their superior nonlinear capability, while PSO is a computationally straightforward and heuristic swarm intelligent algorithm. The ANN, PSO, and PSO-ANN model developed in this work are described progressively.

#### 3.3.1. ANN

Among the various machine learning techniques, ANNs are utilized widely for predicting the adsorption capacity of heavy metals. An ANN can be viewed as a data-processing tool or black box that produces the appropriate output based on the given inputs [7]. A back-propagation network, as the most popular network structure, continuously updates the weights and biases to decrease the errors to an acceptable level [29]. The input layer, hidden layer, and output layer making up the back-propagation neural network have weights, biases, and activation functions. The topological structure of the back-propagation neural network is detailed in Figure 8, where *x_i_* denotes the input variables, $w_{ij}$ represents the weight, and *b* denotes the bias.

The expressions for *d* and the neural network output y are shown in Equations (3) and (4) [30].
(3)$$d=\sum w_{ij}x_{i}+b$$
(4)y=f(d)

The function of bias is to modify the influence of the activation function. The activation function, abbreviated as *f*, enables the neural network to approximate any nonlinear function and is thus applicable to various nonlinear problems. The most widely used activation functions for neural networks are Sigmoid, SoftPlus, tanH, exponential linear unit (ELU), and rectified linear unit (ReLU) [31].

The weights and biases of the ANN model can be adjusted based on the back-propagation errors, as expressed [30]:(5)$$w(t+1)=w(t)-\alpha \frac{\partial E(t)}{\partial w(t)}$$
(6)$$b(t+1)=b(t)-\alpha \frac{\partial E(t)}{\partial b(t)}$$
where α is the learning rate, *t* is the *t*-th iteration, and *E* is the network output error, calculated as expressed [30]:(7)$$E(t)=\sqrt{\frac{1}{n}\sum_{i=1}^{n}{\left|e_{i}\right|}^{2}}$$
where *e_i_* indicates the difference between the actual value and the predicted value, while $\frac{\partial E(t)}{\partial w(t)}$ and $\frac{\partial E(t)}{\partial b(t)}$ indicate the gradients of the output errors for each weight and bias, respectively.

The number of hidden layer neurons is an important parameter of ANNs. The trial-and-error method was used to determine the optimal number of neurons, as reported in our previous work [7]. To unify the magnitudes and improve the speed and accuracy of the model training, the training and testing datasets were normalized. The specific expression for data normalization is described in our previous work [7].

The adaptive learning rate gradient descent, gradient descent, momentum gradient descent, conjugate gradient, and Levenberg–Marquardt (LM) algorithms are often-employed algorithms in the training of ANNs [32]. The selection of an algorithm depends on the situation. As a popular and classic algorithm with a quick convergence and good precision, the LM algorithm was thus selected for this study.

#### 3.3.2. PSO

PSO is an iterative algorithm based on the swarm evolutionary algorithm [33]. The searching process of each particle is simulated in a manner similar to the optimal route of a bird finding food. Therefore, each particle represents a potential solution to a problem. The velocity and position, as the only two properties of each particle, are continuously adjusted by iterations, until an optimal solution is found [34].

Initializing a swarm of random particles is the first step of the PSO algorithm. Assuming that *N* particles make up the population in an *l*-dimensional searching space, the position (*X_m_*) and velocity (*V_m_*) of the *m*-th particle are expressed as two *l*-dimensional vectors.
(8)$$X_{m}=(x_{m1},x_{m2},x_{m3}\ldots x_{ml})\;\;m=1,2,\ldots ,N$$
(9)$$V_{m}=(v_{m1},v_{m2},v_{m3}\ldots v_{ml})\;\;m=1,2,\ldots ,N$$

The optimal position of the *m*-th particle is the individual extreme:(10)$$P_{best}=(p_{m1},p_{m2},p_{m3}\ldots p_{ml})\;\;m=1,2,\ldots ,N$$

The optimal position of the whole population is the global extreme:(11)$$G_{best}=(p_{g1},p_{g2},p_{g3}\ldots p_{gl})$$

When searching for the individual and global extremes, the particle updates its position and velocity as follows:(12)$$v_{ml}^{t}=\omega v_{ml}^{t-1}+c_{1}r_{1}(p_{ml}-x_{ml}^{t-1})+c_{2}r_{2}(p_{gl}-x_{ml}^{t-1})$$
(13)$$x_{ml}^{t}=v_{ml}^{t}+x_{ml}^{t-1}$$
where *r*_1_ and *r*_2_ are randomly selected from the range from 0 to 1 to increase the random searchability. The individual learning factor, *c*_1_, signifies the ability of a particle to search for the best option. The global learning factor, *c*_2_, represents the searchability of the group for the best solution. The default value for the learning factors *c*_1_ and *c*_2_ is 2. The particle is thought to have only one global learning capability if *c*_1_ = 0; at this point, the particle is capable of extended search, but not local search, and converges slowly for difficult issues. The particle is thought to possess only one cognitive capacity if *c*_2_ = 0; at this point, it behaves like a blind random search and is more likely to run into the local optimal solution problem [35].

The inertia factor ω indicates the ability of a particle to continue the previous velocity, which is a fixed value in the conventional PSO algorithm [27]. Its numerical value is positively related to the global search, and inversely related to the local search [36]. Therefore, the global and local search speeds of the PSO algorithm can be enhanced by adjusting the inertia factor. Herein, the linear decreasing inertia factor was adopted to balance the global and local search capabilities of the PSO [37]. This pattern of increasing iterations and decreasing inertia factors ensures both a stronger initial search capability and, subsequentially, a more accurate local search. The specific expression is as follows:(14)$$\omega =\omega_{\max}-\frac{t\times (\omega_{\max}-\omega_{\min})}{t_{\max}}$$
where ω_max_ ∈ [0, 1] and ω_min_ ∈ [0, 1] are the maximum and minimum inertia factors, while *t* and *t*_max_ are the current and maximum number of iterations, respectively. 

#### 3.3.3. PSO-ANN

A flow chart of the PSO-ANN is detailed in Figure 9. The procedure for the prediction of the Cu(II) adsorption capacity of MPP based on the PSO-ANN is as follows:(1)Data Normalization. The training and testing sets were constructed after reading the sample data.(2)Setting the Model Parameters. The maximum number of iterations, neuron numbers of each layer, functions, and termination criteria for the ANN topology were set in accordance with the features of the input data.(3)Optimization of Weights and Biases. The parameters of the PSO were set. The initial weights and biases of the ANN training model were then optimized using the particle swarm algorithm.(4)Establishment of the PSO-ANN model. The optimized weights and biases obtained from the position information of the optimal particle were assigned to the ANN as the initial values for the following training. The testing dataset was applied to verify the well-trained hybrid ANN and output the prediction result.

**Figure 9 molecules-28-06957-f009:**
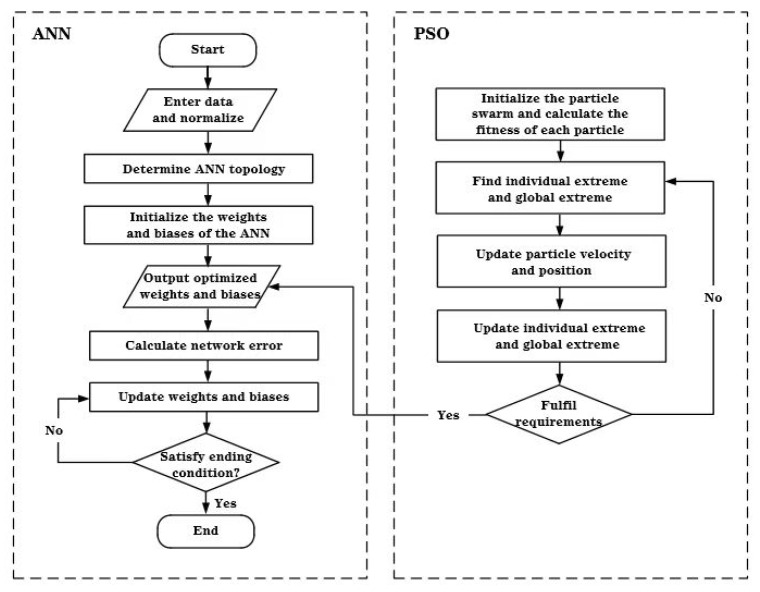
Framework of PSO-ANN.

### 3.4. Model Performance

In the current study, the mean square error (MSE), root mean square error (RMSE), mean absolute error (MAE), and coefficient of determination (*R*^2^) were the statistical metrics utilized to assess the performances of the neural network models. *R*^2^ is widely used to assess model reliability, while the RMSE and MAE reflect the degree of model deviation. These evaluation indicators can be calculated as expressed [38,39]:(15)$$R^{2}=\frac{\sum_{k=1}^{n}\left(y_{0,k}-\bar{y_{0}})(y_{k}-\bar{y}\right)}{\sqrt{\sum_{k=1}^{n}{\left(y_{0,k}-\bar{y_{0}}\right)}^{2}\sum_{k=1}^{n}{\left(y_{k}-\bar{y}\right)}^{2}}}$$
(16)$$MAE=\frac{\sum_{k=1}^{n}\left|y_{0,k}-y_{k}\right|}{n}$$
(17)$$MSE=\frac{1}{n}\sum_{k=1}^{n}{{(y}_{0,k}-y_{k})}^{2}$$
(18)$$RMSE=\sqrt{\frac{1}{n}\sum_{k=1}^{n}{{(y}_{0,k}-y_{k})}^{2}}$$
where *n* is the total number of sample data, y is the output value, $\bar{y}$ is the average output value, $y_{0}$ is the actual value, and $\bar{y_{0}}$ is the average actual value.

## 4. Conclusions

For this study, a PSO-ANN model was proposed to forecast the potential of MPP to adsorb Cu(II). Prior to the model training, a trustworthy database was established. The Pearson correlation coefficient was used to check the redundancy between the input variables. The sensitivity of each input variable on the output variable was calculated using the CAM. A traditional ANN was also established for a comparative analysis. The *R*^2^, MSE, RMSE, and MAE were utilized as assessment indices for describing the overall model performance. According to the training, testing, and combined data results, both the ANN and PSO-ANN demonstrated good predictive abilities, but the proposed PSO-ANN exhibited a greater prediction accuracy. Moreover, the predicted values of the PSO-ANN were closer to the actual values when comparing the model errors. This meant that the PSO effectively compensated for defects in the ANN. In conclusion, this study effectively demonstrated the enhanced validity and accuracy of the PSO-ANN in comparison to the ANN. The PSO-ANN hybrid algorithm integrated and combined the benefits of both single models used. There is great potential for the PSO-ANN to be applied for predicting the adsorption capacity of heavy metals by agricultural waste biosorbents.

## Figures and Tables

**Figure 1 molecules-28-06957-f001:**
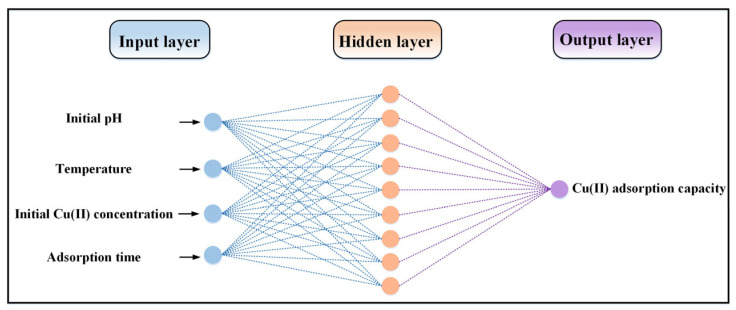
ANN topological structure developed in this study.

**Figure 2 molecules-28-06957-f002:**
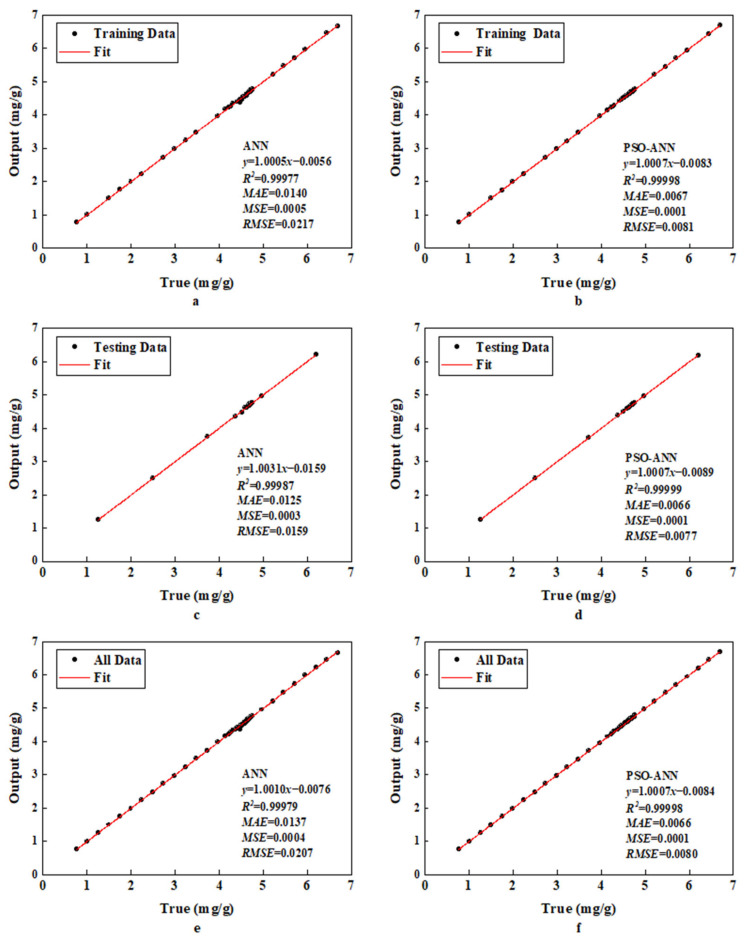
Regression analysis of (**a**,**c**,**e**) ANN and (**b**,**d**,**f**) PSO-ANN for training data, testing data, and all data, respectively.

**Figure 3 molecules-28-06957-f003:**
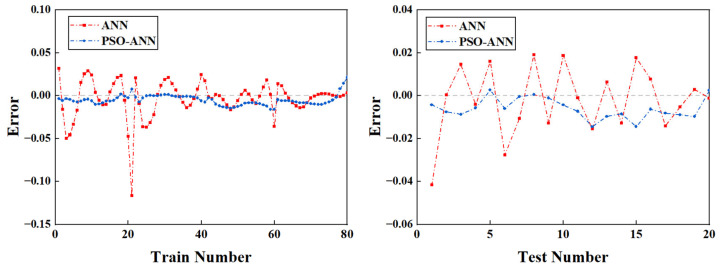
Predicted errors of training (**right**) and testing (**left**) data from the actual values.

**Figure 4 molecules-28-06957-f004:**
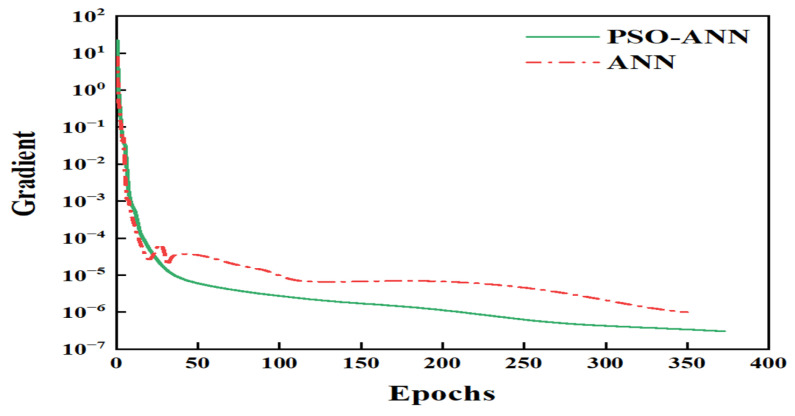
Gradient changes observed for tested models during each iterative training.

**Figure 5 molecules-28-06957-f005:**
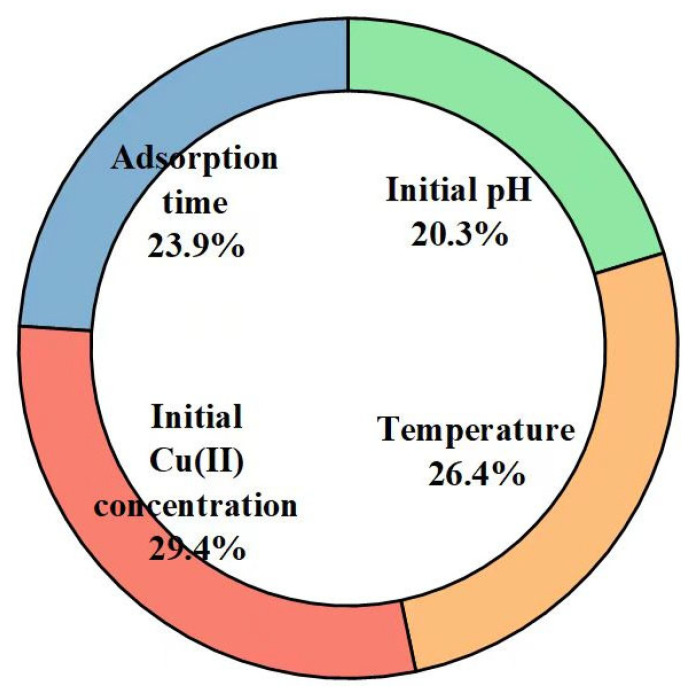
Relative importance of input variables based on the PSO-ANN weight matrix.

**Figure 6 molecules-28-06957-f006:**
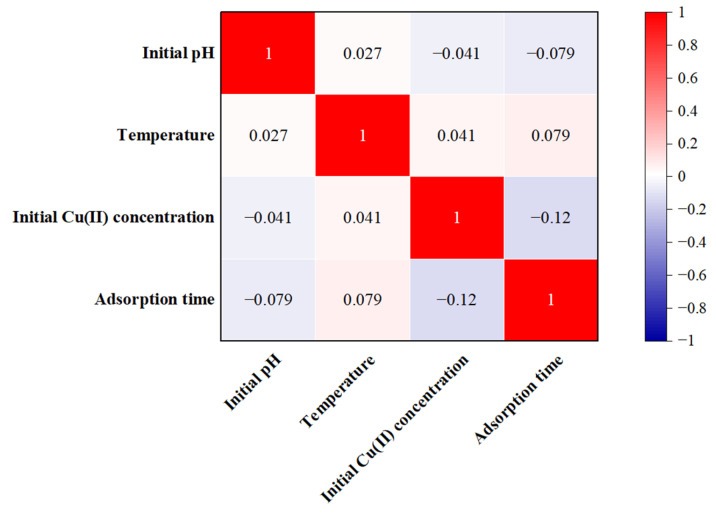
Heat map of the Pearson correlation.

**Figure 7 molecules-28-06957-f007:**
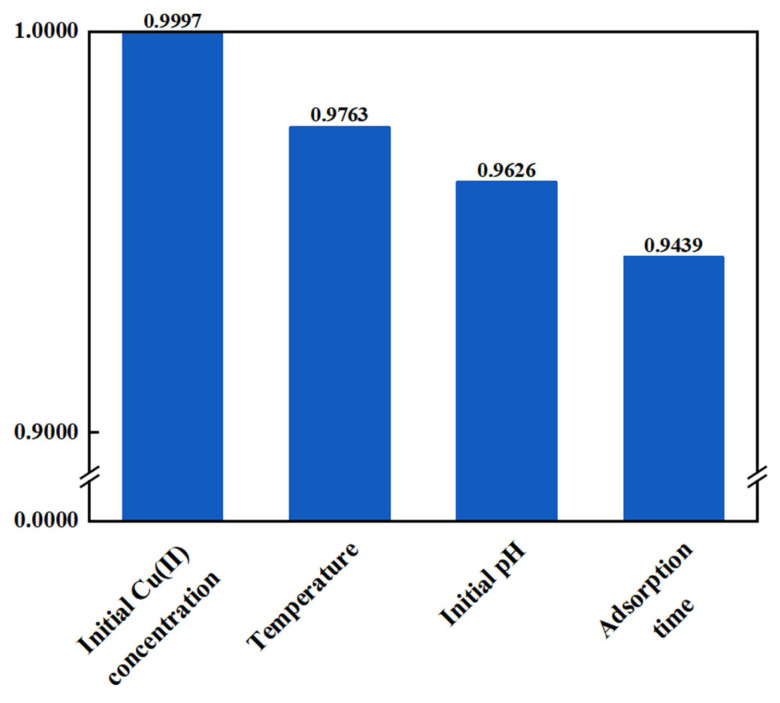
Sensitivity factors of input variables.

**Figure 8 molecules-28-06957-f008:**
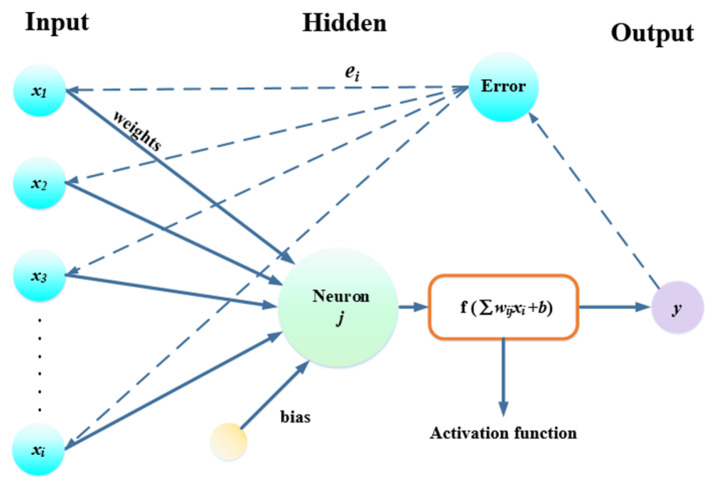
Structure of the back-propagation neural network.

**Table 1 molecules-28-06957-t001:** Model parameters of the ANN.

Parameter	Setting
Input layer node	4
Hidden layer node	9
Output layer node	1
Activation function	tansig, purelin
Training function	trainlm
Epochs	1000
Learning rate	0.01
Precision	1 × 10^−5^
Epochs between display	25
Momentum factor	0.01
Minimum performance gradient	1.00 × 10^−6^
Maximum validation failure	6

**Table 2 molecules-28-06957-t002:** Performance evaluation of ANN and PSO-ANN.

		ANN	PSO-ANN
*R* ^2^	Training	0.99977	0.99998
	Testing	0.99987	0.99999
MSE	Training	0.0005	0.0001
	Testing	0.0003	0.0001
RMSE	Training	0.0217	0.0081
	Testing	0.0159	0.0077
MAE	Training	0.0140	0.0067
	Testing	0.0125	0.0066

**Table 3 molecules-28-06957-t003:** Model parameters of PSO-ANN.

Parameter	Setting
Particle swarm size	20
Maximum iterations	60
*c* _1_	2
*c* _1_	2
*ω* _max_	0.90
*ω* _min_	0.40
Position constraint	[−5, 5]
Velocity constraint	[−1, 1]

**Table 4 molecules-28-06957-t004:** Comparison with other ANN-based hybrid models.

Reference	Model	Removal Ions	Dataset	Performance	
				Training	Testing
This study	PSO-ANN	Cu(II)	100	*R*^2^ = 0.99998 RMSE = 0.0081 MAE = 0.0067	*R*^2^ = 0.99999 RMSE = 0.0077 MAE = 0.0066
Zheng et al. [20]	QSA-ANN	Cu(II), Pb(II), Zn(II), As(III), Cd(II), and Ni(II)	353	*R*^2^ = 0.978 RMSE = 0.051	*R*^2^ = 0.960 RMSE = 0.074
Ke et al. [21]	SVM-ANN	Cu(II), Pb(II), Zn(II), As(III), Cd(II), and Ni(II)	353	*R*^2^ = 0.995 RMSE = 0.036 MAE = 0.018	*R*^2^ = 0.987 RMSE = 0.046 MAE = 0.026
Bhagat et al. [22]	ANN-M5	Cu(II)	95	*R*^2^ = 0.9983 RMSE = 1.3799 MAE = 1.0338	*R*^2^ = 0.9974 RMSE = 0.9283 MAE = 0.6200
Pooladi et al. [23]	GMDH-ANN	Pb(II)	Not reported	*R*^2^ = 0.94 RMSE = 3.5524 MAE = 2.6958	*R*^2^ = 0.9315 RMSE = 5.3083 MAE = 4.041
Zafar et al. [24]	ANFIS	Cr(VI)	18	*R*^2^ = 0.99 RMSE = 0.63	*R*^2^ = 0.94 RMSE = 6.23

**Table 5 molecules-28-06957-t005:** Statistical analysis of each variable.

	Max	Min	Average	Median	Standard Deviation	Skewness
Temperature (K)	313.15	288.15	298.80	298.15	3.98	1.33
Initial pH	7.00	2.00	4.87	5.00	0.80	−1.33
Adsorption time (min)	60.00	10.00	53.50	60.00	13.37	−1.99
Initial Cu(II) concentration (mg/L)	28.00	4.00	19.00	20.00	4.00	−1.95
Cu(II) adsorption capacity (mg/g)	6.6994	0.7726	4.4236	4.6865	0.9820	−1.8389

## Data Availability

Data are contained within the article or the Appendix A.

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
