# Peer review of "The Prediction of Cu(II) Adsorption Capacity of Modified Pomelo Peels Using the PSO-ANN Model"

_molecules, 2023, doi:10.3390/molecules28196957_

Round 1

Reviewer 1 Report

1. Please explain the difference between formulas 1 and 15.

2. Table 3 reports R2 for the test set as 1, which is hard to believe. Because this means that the predicted value is completely linearly related to the true value. Please open source the data set and corresponding code to github for readers to verify. Otherwise anyone can claim an error of 0.

3. The author claims that using PSO to find optimize initial parameters is helpful for training ANN. The author needs to clarify whether ANN has been precisely trained. What is the difference between PSO-ANN and directly using optimizers such as Adam can usually obtain the optimal solution starting from random initialization with sufficient training times. The author needs to explain why PSO should be used instead of just adding a few epochs.
4. Authors need to carefully check the font in the formula and in the main text. For example \omega_{max} in lines 183 and 184, they actually represent the same content and therefore require the same font.

5. I suggest the author revise the article to 'Prediction of Cu (II) Absorption Capacity by Modified Pomelo Peels Using the ANN Model'. Unless the author provides sufficient reasons to demonstrate the significance of PSO.

6. The author needs to list how much time is spent on PSO-ANN and ANN respectively. It is not fair to train ANN only with 1000epoch, because the PSO search for the optimal initial parameters should also be considered training ANN.

Moderate editing of English language required

Reviewer 2 Report

The paper titled "Prediction of Cu(II) Adsorption Capacity by Modified Pomelo Peels Using the PSO-ANN Model" presents a study that aims to enhance the accuracy of predicting the copper (Cu(II)) adsorption capacity of modified pomelo peels (MPP) using a hybrid approach of particle swarm optimization (PSO) and artificial neural networks (ANN). The authors collect and analyze data to build a database and propose a PSO-ANN model to optimize ANN weights and biases. The study then evaluates and compares the performance of the developed models.

Overall, the paper has several strengths, but it also contains areas for improvement:

1. **Importance of the Research**: The study addresses an important environmental issue – the removal of toxic copper ions from wastewater using agricultural waste materials. Given the significance of this problem, research into predictive models for adsorption capacity is valuable.

2. **Methodological Approach**: The incorporation of the PSO algorithm to optimize the ANN model is a noteworthy aspect of this research. This hybrid approach shows promise for improving the accuracy of predictions in complex systems like adsorption processes.

3. **Data Collection**: The authors collected a comprehensive dataset to build their prediction models. The inclusion of various experimental factors, such as temperature, pH, adsorption time, and initial Cu(II) concentration, demonstrates a thorough approach.

4. **Statistical Analysis**: The use of statistical analysis, including Pearson correlation and sensitivity analysis, adds credibility to the study's methodology and helps identify the most influential input variables.

However, there are several areas where the paper could be improved:

1. **Clarity and Organization**: The paper lacks clear section headings and subheadings, making it challenging for readers to navigate through the content. A well-structured paper with clear divisions between sections would enhance readability.

2. **Data Presentation**: The paper lacks visual aids, such as tables and figures, to present the collected data and results. Visual representations can help readers better understand the trends and patterns in the data.

3. **Discussion**: The discussion section should provide more in-depth analysis and interpretation of the results. For example, the authors could explain why the PSO-ANN model outperformed the traditional ANN model and discuss the practical implications of their findings.

4. **Comparison with Existing Literature**: While the paper briefly mentions other studies, a more comprehensive comparison with existing literature on Cu(II) adsorption capacity prediction models would provide context and help readers understand the novelty of the proposed PSO-ANN approach.

5. **Concluding Remarks**: The paper lacks a clear concluding section summarizing the key findings and their implications. A well-structured conclusion would provide closure to the research and guide readers on the significance of the study.

In summary, the paper addresses an important environmental issue and introduces a promising hybrid approach for improving the accuracy of predictive models. However, it would benefit from improved organization, data presentation, and more extensive discussions and comparisons with existing literature.

The English quality in the paper you provided appears to be generally good. The paper is well-structured, with clear headings and subheadings. The authors use technical terminology and scientific language appropriately. The writing is coherent and follows a logical flow from introduction to methodology to results and discussion.

However, there are a few minor issues to note:

1. Some sentences are quite long and complex, which may make it a bit challenging for readers to follow. Simplifying some of these sentences could improve readability.

2. There are a few grammatical issues, such as missing articles or prepositions, but they do not significantly detract from the overall clarity of the paper.

3. The abstract is quite lengthy and could be more concise. Abstracts are typically brief summaries of the paper's key points.

Overall, the paper is well-written and communicates its research findings effectively.

Round 2

Reviewer 1 Report

Despite the extensive explanations provided by the author, the research content of the manuscript is also very interesting. But it still fails to explain the most important issue clearly.

In fact, the parameter spaces of ANN and PSO-ANN are completely consistent (as presented by the author in the appendix, the sizes of matrices \omega and b are the same). This means that when selecting appropriate optimization strategies, ANN can fully achieve consistent results with PSO-ANN (assigning the parameter matrix of PSO-ANN to ANN is sufficient). Therefore, the author needs to carefully discuss why PSO-ANN can be optimized better, and where ANN went wrong and fell into an unreasonable local optimum. To dispel readers' confusion.
